rsos.royalsocietypublishing.org

materials science/crystallography

cyclic dimer, $\alpha$- and $\beta$-form, characteristic, transition, sublimation, melting

**Author for correspondence:**
Chunwang Yi
e-mail: cwyi@hunnu.edu.cn

This article has been edited by the Royal Society of Chemistry, including the commissioning, peer review process and editorial aspects up to the point of acceptance.

†Lu Peng and Jie Li are co-first authors.

# The crystal-form transition behaviours and morphology changes in a polyamide 6 cyclic dimer

Lu Peng[1,†], Jie Li[3,†], Shumin Peng[1], Chunwang Yi[1,2] and Feng Jiang[4]

[1]Key Laboratory of Sustainable Resources Processing and Advanced Materials of Hunan Province, and [2]National and Local Joint Engineering Laboratory for New Petro-chemical Materials and Fine Utilization of Resources, Hunan Normal University, Changsha 410081, Hunan, People's Republic of China
[3]College of Chemistry and Materials Science, Hengyang Normal University, Hengyang 421008, Hunan, People's Republic of China
[4]State Key Laboratory of Biobased Fiber Manufacturing Technology, China Textile Academy, Beijing 100025, People's Republic of China

CY, 0000-0003-4398-8058

The characteristics of pure $\alpha$- and $\beta$-form of the cyclic dimer (1,8-diazacyclotetradecane-2,9-dione) were systematically and integrally investigated during this study. The results showed that the $\alpha$-form could dissolve and rapidly transform into the $\beta$-form in methanol, and in caprolactam solution at a lower temperature, an interesting transition occurred and formed co-precipitates, which refract colourful light under PLM. However, these dimers can aggregate in water, and they are then transformed into multi-slice layers and compact structures. The detailed transition behaviours between the two forms were further measured by FT-IR, XRD and DSC by varying the temperature from 25°C to 360°C, respectively, which showed that there are two endothermic transitions over the course of the heating programme. At a temperature of approximately 242°C, the $\beta$ crystals were initially converted into $\alpha$ crystals, and then they melted when the temperature reached over 345°C. A video recorded under a light microscope also showed that the sublimation of the $\beta$ cyclic dimer occurred after the transition. However, the $\alpha$-form might sublimate at temperatures lower than 150°C when mixed with volatile matter.

## 1. Introduction

Polyamide 6 (PA6), which is known as nylon 6, is widely used in engineering thermoplastics and fibres due to its excellent chemical stability and mechanical strength. At present,

rsos.royalsocietypublishing.org   R. Soc. open sci. **5**: 180957

hydrolysing caprolactam by using water as initiator remains the only way to produce PA6 at a large scale [1]. Corresponding chemical mechanisms have disclosed that approximately 11% water-soluble materials, also called extractables include monomers, cyclic dimers, cyclic trimers and other low-molecular-weight oligomers, were unavoidably generated in the polymer when the caprolactam polymerization achieved equilibrium.

Cyclic dimers are a very important and special form of PA6 extractables, and they were first recognized by I. G. Farben and K. Hoshino and were recorded in their earliest brief notes [2]. I. G. Farben and K. Hoshino also found that although most oligomers were removed by hot desalted water before further processing, there was still a small amount of cyclic dimer remaining in the PA6, which always led to serious damage in the downstream processing and product quality [3]. More detailed investigations were performed by Hermans [4,5], who isolated two isomers of cyclic dimers from PA6 and designated the needle-shaped crystal with a melting point of 247°C as the $\alpha$ cyclic dimer, while another flaky crystal was named the $\beta$ cyclic dimer. I. Rothe & M. Rothe [6] asserted that the two forms were actually a dimer and trimer, and later X-ray investigations performed by Northeolt & Alexander [7,8] did not account for the presence of these two forms. This ambiguity was resolved through the agreement by Hermans.

In the decades to follow, Heikens [9] first attempted to determine the cyclic dimer and the other cyclo-oligoamides up to the tetramer in a quantitative form. M. Rothe [10] succeeded in performing a quantitative determination of cyclo-oligoamide 6 up to the monomer by paper chromatography. Later, Anton used the infrared spectrum to determine the cyclic oligomer contents quantitatively [11]. Until 1970, a gel permeation chromatogram was used to determine the contents of mixtures consisting of cyclic dimers to hexamers [12]. In 1974, Andrews *et al*. [13] reported a method for separating the cyclic dimer and monomer of nylon 6 on an OV-1 type column using a linear temperature programme. In 1977 and 1979, caprolactam and oligomers were determined by LPC and MS for the first time [14,15]. Following that study, Krajnik *et al*. [16] and Tai & Tagawa [17] used reverse-phase HPLC to separate cyclic oligomers with water/methanol mixtures as the eluents. However, it is impossible to introduce an internal standard, which is essential for reproducible, quantitative results. Guaita [18] further developed HPLC technology in 1984 by using trifluoroethanol/water (60 : 40) as eluents, and 1–10 cyclic oligomers could be separated and determined. In general, by using chromatographic techniques, single oligomers have been resolved well and clearly identified.

However, the most frequently reported technologies were aimed at separating and determining the cyclic oligomers, and fewer of them focused on the material characteristics, especially the sublimation and transition behaviours between the $\alpha$ and $\beta$-forms. Di Silvestro *et al*. [19] disclosed the chemical structure of the $\alpha$- and $\beta$-forms and further investigated the characteristics of the cyclic dimers. By using DSC, they found that the $\beta$-form cyclic dimer obtained after crystallization from protic solvents presented two endothermic transitions at 244°C and 347°C. However, the $\alpha$ cyclic dimer prepared by sublimation at 210°C (2 mmHg) showed only one transition at 347°C. Di Silvestro *et al*. posited that 244°C is the transition temperature rather than the melting point of the $\beta$ cyclic dimer as assumed by Hermans. Kržan [20] also found an additional minor peak at 235°C on the first heating scan curve of DSC, which corresponds to the transition temperature from the $\beta$ crystal form to the more stable $\alpha$ crystal structure, and it was absent from the second heating scan following the cooling (10 K min$^{-1}$) of the melted sample. Furthermore, Kržan noted that the effect of the temperature increase is most noted in the N–H stretching peak that shifts to higher wavenumbers.

Beyond that, some papers on the formation, recycling and polymerization of the cyclic dimer were published [21–24]. In recent years, caprolactam ring-opening hydrolytic polymerization technology has developed very quickly and has inspired a surge in the output of polyamide 6. Large amounts of PA6 and PA6-based copolymers have been used to produce food packages, medical glue, baby clothes and other medical and personal care products [25] for which safety is in high demand due to the special service environment. Hence, the migration and precipitation behaviours of oligomers, together with biodegradation, have drawn more researchers to focus on this material [26–31]. Moreover, in industrial PA6 plants, especially in monomer recovery unit, the $\beta$-form cyclic dimer in concentrate (also named water-soluble extractable with higher concentration) can easily arouse great aggregation in equipment and pipes.

Although previous literature showed the structure of the cyclic dimer, the most noteworthy features of its two forms, their morphological changes under different conditions (solvents, air, high temperature and others) and specific transition behaviours were not clear. Particularly, the thermal stability and the reason for dimer agglomeration have not been revealed yet. These properties prompted us to employ modern techniques to investigate the distinctive features of the cyclic dimer in a systematic and integral form, including their morphological change in methanol, water and caprolactam, before and after sublimation at a high temperature. In particular, the entire transition between the $\alpha$ and $\beta$ forms was observed by

rsos.royalsocietypublishing.org R. Soc. open sci. 5: 180957

XRD and FT-IR (with a temperature change accessory) united microscopy for the first time; moreover, the transition temperature and melting points of the $\alpha$ and $\beta$ cyclic dimer forms were identified using DSC and Schlenk tubes models. Some new findings were discovered, and they are reported in this paper. The results may provide valuable theoretical guidance for further investigation into the more serious problem caused by the aggregation of cyclic dimers in modern industrial PA6 plants.

# 2. Material and methods

## 2.1. Preparation and purification of the cyclic dimer

An oligomer residue was provided by an industrial PA6 producer (Jinjiang Technology Co. Ltd, Fujian, China). By sublimating the oligomer residue at 180°C and $10^{-3}$ Torr to remove the other oligomers in accordance with the literature, the needle-shaped $\alpha$-form cyclic dimers were isolated. Following that step, the needle-like crystals were cooled and dissolved in 10 times the volume of methanol, and then with the evaporation of the solvent, white $\beta$-form slices emerged in the volumetric flask. The dissolution and evaporation processes were repeated three times to eliminate minute impurities, and finally, a high purity $\beta$-form cyclic dimer (99.9%) was obtained. Through the repeated sublimation of the high purity $\beta$-form cyclic dimer at 180°C and $10^{-3}$ Torr, a white $\alpha$-form cyclic dimer with higher purity was collected and sealed in a vacuum bottle for further study.

## 2.2. Measurement and characterization

$^{1}$H NMR and $^{13}$C NMR were performed using a Bruker Avance-500 at 500 Hz NMR spectrometer at room temperature. CDCl$_3$ was used as a solvent.

The reflection method was adopted to analyse the Fourier transform infrared (FT-IR) spectra for cyclic dimers over a range of $4000–500$ cm$^{-1}$, and the data were recorded by Nicolet iS10 FT-IR spectrometer with a scanning accuracy of $4$ cm$^{-1}$.

Variable Fourier transform infrared spectra were tested by the transmittance method. Signals ranging from 4000 to 400 cm$^{-1}$ were recorded by a Bruker Equino55 FT-IR spectrometer at 25–360°C using a heating speed of 5°C min$^{-1}$ and a scanning accuracy of 4 cm$^{-1}$.

Synchrotron X-ray measurements were performed using a Shanghai synchrotron, and the radiation two-dimensional (2D) WAXD pattern was acquired using a Mar-CCD (345) detector. The sample-to-detector distance for the WAXD was 277.86 mm. The XRD experiments were performed on a PANalytical X'Pert Pro MPD X-ray diffractometer with nickel-filtered Cu-K$\alpha$ radiation ($\lambda = 1.54$ Å) at 45 kV and 40 mA. X-ray scanning was performed over a range of $5° < 2\theta < 50°$ with a scan rate of $5°(2\theta)$ min$^{-1}$. By varying the scan rate to $35°(2\theta)$ min$^{-1}$, the variable spectra of the samples were measured at a heating rate of 2°C min$^{-1}$ from 25°C to 280°C.

The transition and melting behaviours of the cyclic dimers were recorded with a Q20 differential scanning calorimeter under a nitrogen atmosphere. The scan process was run from 30°C to 360°C at heating speeds of 10 and 20°C min$^{-1}$, respectively.

The transitions between the $\beta$-form and $\alpha$-form in water, methanol and liquid caprolactam were observed under a NOVEL/N-180M microscope equipped with a hot stage during the heating process. The transition process was recorded by the video.

The scanning electron microscope (SEM Hitachi S-4300) was adopted to observe the morphology of the cyclic dimers. The samples used for testing were cleaned and spray-coated with a thin layer of evaporated gold to examine the micrographs.

$\beta$-form cyclic dimer powder was loaded into Schlenk tubes, which were then sealed under a vacuum. Following that step, the tubes were heated at the designated temperature in a muffle furnace. They were then cooled to room temperature and photographed.

# 3. Results and discussion

## 3.1. Original morphologies

Cyclic dimers are generally generated by the back-biting of the amino end groups over the course of the amide exchange shown as scheme 1, and it is clear that there are two amide groups in the cyclic molecule.

rsos.royalsocietypublishing.org    R. Soc. open sci. **5**: 180957

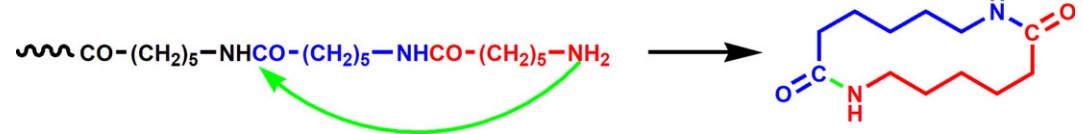

**Scheme 1.** Formation of the caprolactam cyclic dimer and its molecular structure.

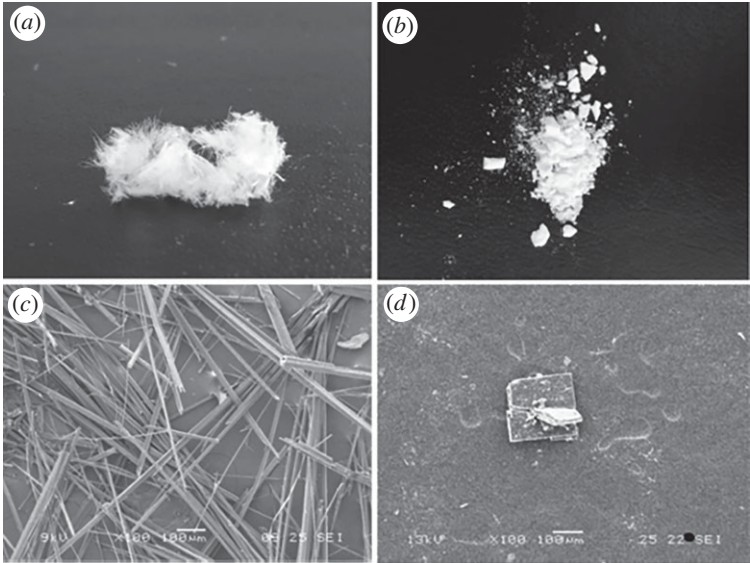

**Figure 1.** Morphology of cyclic dimer: (*a*) α-form; (*b*) β-form; (*c*) SEM-magnified image of α-form; and (*d*) SEM-magnified image of β-form.

The morphologies of the α and β cyclic dimers were photographed by digital camera and are presented in figure 1. The α-form crystal obtained at a high temperature is shown in figure 1*a*, and it looks like a fluffy thread or regular needles under SEM at a magnification of 100 times (figure 1*c*). The white powders indicated in figure 1*b* are the β-form prepared in methanol, and they are actually composed of many rectangular plates (figure 1*d*) as shown when enlarged 100 times by SEM.

## 3.2. Characteristics

[1]H NMR and [13]C NMR were performed to uncover the structure of the cyclic dimer, and the caprolactam signals were also caught to help us to identify the variations in chemical shifts. The curves are displayed in figure 2. The relevant assignments and chemical shifts of proteins obtained by [1]H NMR and [13]C NMR are given in table 1. $\delta$ (ppm) for [1]H NMR, 2.92 (e, $CH_2$-NH), 2.05 (f, $CH_2$-CO), 1.14 ppm (g, -$CH_2$-), 1.02 (h, -$CH_2$-), and 0.68 (I, -$CH_2$-); for [13]C NMR, the shifts at 174.15 ppm belong to the carbon atom of the carbonyl (g), where it appears at 36.87 ppm and is attributed to the methane adjacent to the amino-group (h). The peak of 28.53 ppm is attached to the methane close to the carbonyl (i), and the shifts of 21.64, 19.75 and 19.48 ppm correspond to the carbon atom in methane for which the positions are k, l and j in the ring of the cyclic dimer, respectively.

The cyclic dimer was also characterized by Fourier transform infrared spectrometry (FT-IR) and X-ray diffraction (XRD). Figure 3*a* presents the FT-IR spectra of the α-form and β-form, and in the spectrum of the α-form cyclic dimer, the band that appears at 3302 cm$^{-1}$ is attached to the stretching vibration of the amino groups (N–H); 1660–1552 cm$^{-1}$ are related to the stretching vibration of the carbonyl (C=O) groups; and the absorptions of 1470 and 725 cm$^{-1}$ are defined as scissoring vibrations of methane (C–H) and a rocking vibration of methane (C–H, $n > 4$), respectively. Regarding the spectrum of the β-form, the characteristic peaks are similar to those of the α-form. However, the peak located at 2930 cm$^{-1}$ in the α-form vanished, only to be replaced by two new peaks at the 2952 and 2921 cm$^{-1}$ bands, which represent asymmetric and symmetric $CH_2$ stretching modes, respectively. Additionally, the band for the stretching vibration of the amino groups (N–H at 3302 cm$^{-1}$) shifts towards a lower frequency of 3258 cm$^{-1}$.

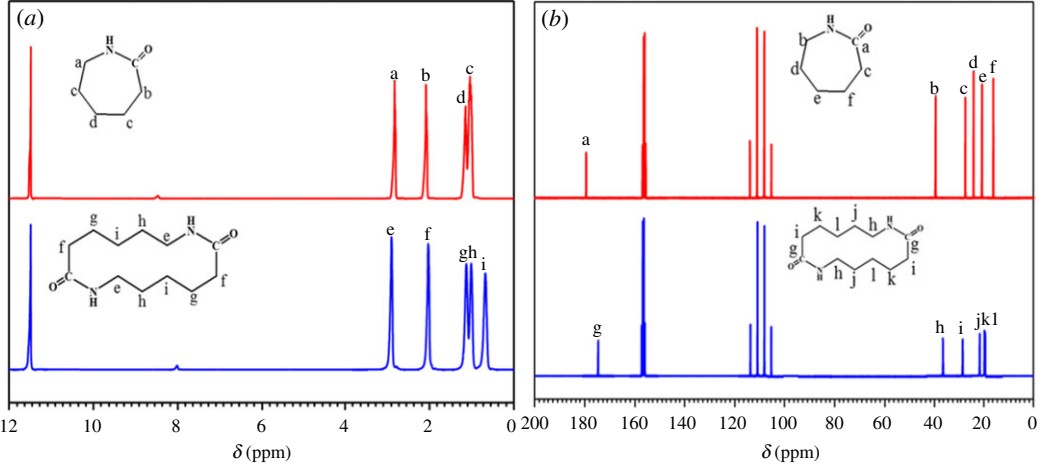

**Figure 2.** Spectra from [1]H NMR and [13]C NMR for the cyclic dimer and caprolactam. (*a*) [1]H NMR and (*b*) [13]C NMR.

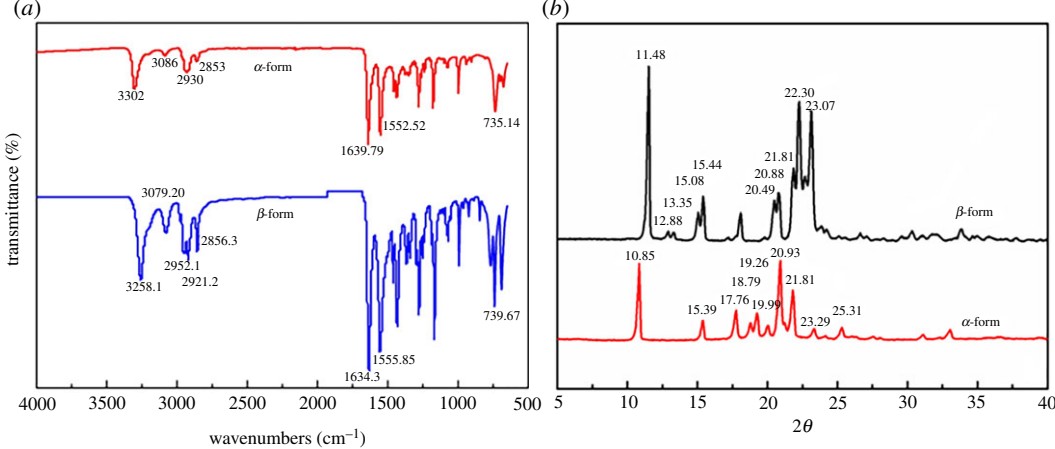

**Figure 3.** FT-IR and XRD curves of the cyclic dimer. (*a*) FT-IR curves; (*b*) XRD curves.

**Table 1.** Assignment of proteins and their shifts.

| structure | cyclic dimer | |
| --- | --- | --- |
| | assignment of protein | chemical shift (ppm) |
| [1]H NMR | $CH_2$-NH (e) | 2.92 |
| | $CH_2$-CO (f) | 2.05 |
| | -$CH_2$- (g) | 1.14 |
| | -$CH_2$- (h) | 1.02 |
| | -$CH_2$- (i) | 0.68 |
| [13]C NMR | -CO- (g) | 174.75 |
| | $CH_2$-NH (h) | 36.87 |
| | -CO- (i) | 28.53 |
| | -$CH_2$- (j) | 21.64 |
| | -$CH_2$- (k) | 19.75 |
| | -$CH_2$- (l) | 19.48 |

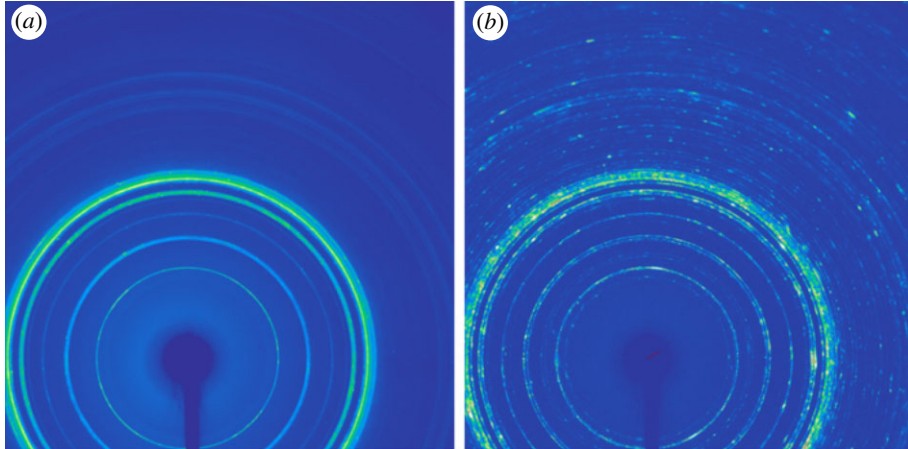

**Figure 4.** WXRD patterns of cyclic dimer. (a) $\alpha$-form and (b) $\beta$-form.

The WXRD patterns of the $\alpha$-form and $\beta$-form cyclic dimer crystals are shown in figure 4 and their X-ray diffraction spectra of the cyclic dimer are displayed in figure 3b. The characteristic peaks of the $\alpha$-form appear at $2\theta$ as follow: $10.85°$, $15.39°$, $17.76°$, $18.79°$, $19.26°$, $19.99°$, $20.93°$, $21.81°$, $23.29°$ and $25.31°$. The diffraction peaks at $11.48°$, $12.88°$, $13.35°$, $15.08°$, $15.44°$, $20.49°$, $20.88°$, $21.81°$, $22.30°$ and $23.07°$ are attached to the $\beta$ crystals. The peaks are very consistent with the reported results given by Di Silvestro et al. [19], who disclosed that this packing rearrangement is connected to an important conformational change along the aliphatic sequence of the dimer. In the centre of the $\beta$-form crystal, hydrogen bonds build up an almost square two-dimensional lattice with triclinic cell dimensions for axes $b$ and $c$ (8.91 Å with an angle of 90°), each cyclic dimer molecule is connected by four hydrogen bonds to four other molecules, which make the crystal more stable at a lower temperature. However, the $\alpha$-form crystal obtained at a higher temperature is much more symmetric. In the centre of the $\alpha$-form crystal, hydrogen bonding gives rise to rows of molecules instead of a two-dimensional network. When viewing the structure along the axis, each ($x$, $y$ and $z$) molecule is linked by hydrogen bonds (2.93(1) Å) exclusively with its ($x \pm 1$, $y$, $z$) neighbours so the axis coincides with the direction of the hydrogen bonding propagation. This structure results in the extremely elongated shapes of the crystals that make them look like thin needles.

## 3.3. Solubility and morphologies in different solvents

In this study, the dissolution and morphology of the $\alpha$- and $\beta$-form cyclic dimers in methanol, caprolactam and water have been investigated using light microscopy under 80 times magnification and SEM with a magnification of 500 times. The results show that both types of cyclic dimer could be fully dissolved in methanol at room temperature. The maximum solubility reaches 22%. Following that step, a drop of cyclic dimer/methanol solution was dropped on the glass substrate to investigate the change in morphology under light microscopy. Interestingly, as the methanol rapidly evaporated over a very short time (less than 10 s in this case), many small but regular rectangles of the $\beta$-form emerged (figure 5a).

Hot caprolactam solution is capable of dissolving a small amount of dimer, and in this study, a maximum solubility of 5% was obtained at 120°C. However, when decreasing the temperature of the caprolactam solution to lower than 80°C, it was amazing to find that many co-precipitates formed and looked like irregular lamellae as shown in figure 5b because the $\beta$ crystals have a high affinity for caprolactam, which could be identified easily by the colourful refracting light under PLM. By increasing the temperature to above 120°C again, the colourful light vanished and the dimer re-dissolved in caprolactam.

It is generally known that cyclic dimers are insoluble in water, Hermans showed that the solubility of the cyclic dimer in water is approximately 0.1% [5]. However, by adding 10 times the volume of water to the $\alpha$-form cyclic dimer, the needle-like crystals transferred to the regular square $\beta$-form very quickly, although they are not soluble in water, as shown in figure 5c. These crystals are larger than those formed in methanol, and they easily aggregate and precipitate in water (figure 5d). A more detailed analysis executed by SEM showed that these aggregates are primarily present as multi-slice (figure 5e) and compact structures (figure 5f). As reported by Di Silvestro et al., each cyclic dimer molecule is connected by four hydrogen bonds to four other molecules. We believe that the exposed molecule of

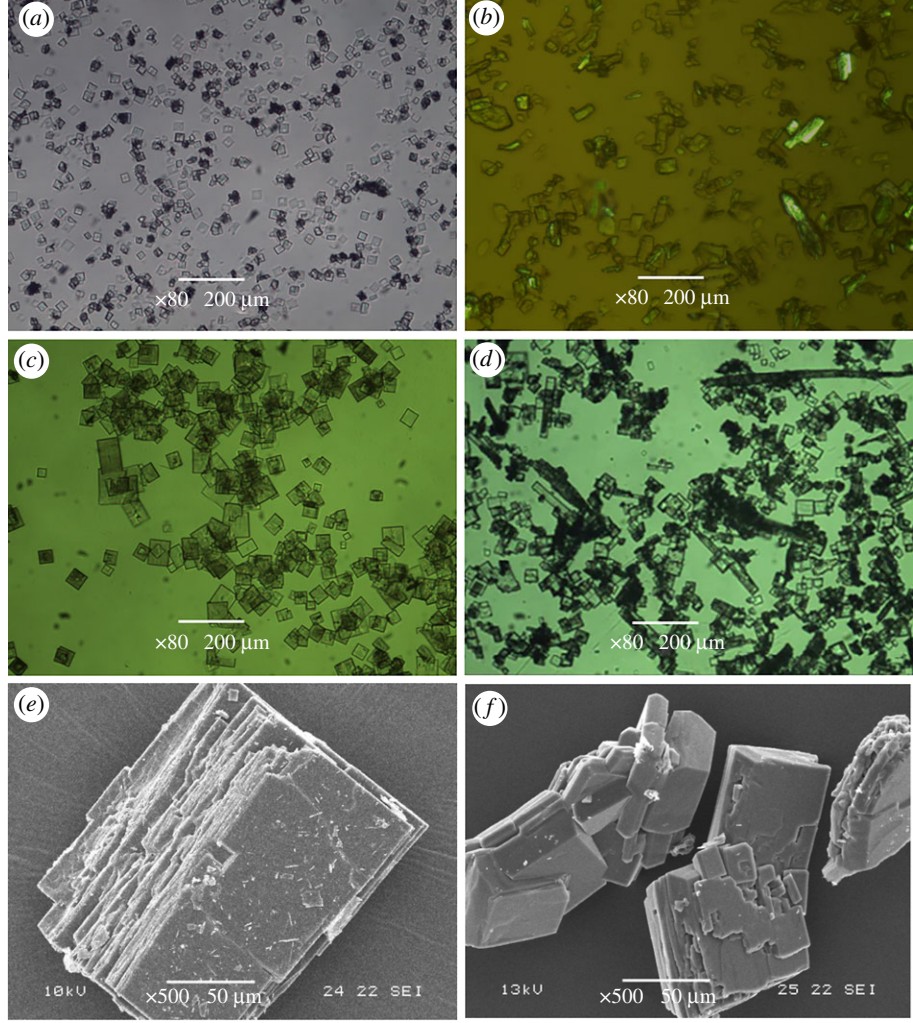

**Figure 5.** Morphologies of cyclic dimers in different solutions. (*a*) In methanol; (*b*) in caprolactam; (*c*) in water; (*d*) aggregates in water; (*e*) slice layers of aggregates; and (*f*) compact structure of aggregates.

cyclic dimer slices could also link to the other slices, caprolactam, other oligomers and even water through hydrogen bonding at relatively low temperatures, thus reconstructing the larger and thicker plates. This bonding could explain how concentrates in PA6 plants with high quantities of cyclic dimers are harmful because they plug up pipelines and equipment. Moreover, the larger and thicker β-form cyclic dimer plates are difficult to remove completely from PA6 in extracting process due to its poor solubility in water, as a result, the remaining β-form cyclic dimer in PA6 would bring quality and safety problems to food packages, medical glue, baby clothes and other medical and life-care products, because they can migrate slowly from the inner region to the surface of polymer. Hence, the detailed technology of the transition of cyclic dimer can greatly help us to take the appropriate solutions and implement controls to ensure the security of PA6 product.

## 3.4. Transition between $\alpha$- and $\beta$-forms

Researchers have tried to investigate the transition between the $\alpha$- and $\beta$-form cyclic dimers in recent decades. However, very little information about the process has been reported in detail. In this study, the transition process has been studied carefully by light microscopy with a heating programme, and the results are shown in figure 6.

As shown as figure 6*a*, 0.5 mg of $\alpha$-form cyclic dimer was spread on the glass substrate at room temperature, and after a drop of water was added, the needle-like crystals quickly transformed into star-like lamellas (figure 6*b*), with the initial shape of the $\beta$-form cyclic dimer first. With the increasing temperature, more and more regular squares appeared and moved around in random patterns

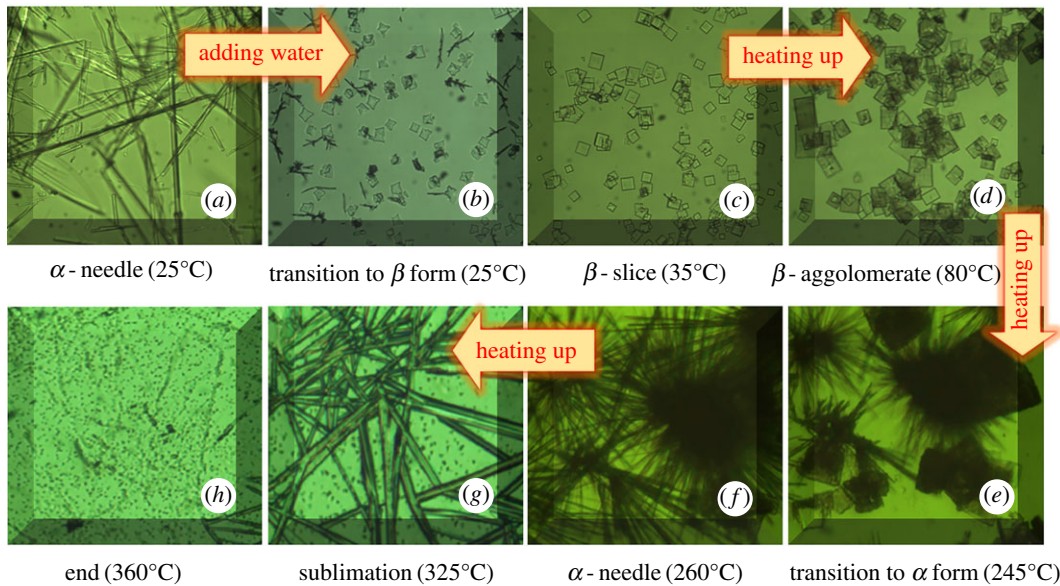

**Figure 6.** Transition of cyclic dimer during the heating programme.

(figure 6c). Finally, they gathered into thicker β slices (figure 6d), with the evaporation of the water. When the temperature increased to over 245°C, bundles of thin needles were observed to be slowly growing up around the thick slices (figure 6e), indicating that the β-form began to transform into the α-form again. In the meantime, the slices moved out of sight gradually. Clearly, the α-form needles that transformed from the β-form, in this case, look black and are thinner than those obtained by sublimating the oligomer residues. At approximately 260°C, almost all of the β-form has transitioned to the α-form (figure 6f), and the entire transition process of the β-form to the α-form occurred over 25 min in this case. Further analysis indicated that the transition time was highly dependent on the mass of the sample and the heating speed. In addition, as the temperature increased, the needles became thicker but transparent, and as the temperature increased continuously to more than 320°C, the sublimation of the α-form was visible, and the needles vanished very rapidly (figure 6g). At last, when the temperature increased to 360°C, all the needle-like α-form cyclic structures disappeared and some liquid dots remained on the glass substrate (figure 6h).

The results also show that under suitable conditions, the α-form and β-form can transform into one another. The β-form is very stable in solvents or at room conditions, but it will transform into the α-form when the temperature is above 245°C. However, the α-form can only remain stable in the absence of solvents, and it sublimates easily at high temperatures. These studies clearly disclose a series of phenomena happened in PA6 spinning process: firstly, the remained β-form cyclic dimer in PA6 transited to α-form during melt-extrusion at a temperature higher than 250°C, and then volatilized when they released from the small holes of spinneret, later, they were quickly cooled to needle-shape α-form cyclic dimer when exposed to the air. These needles were unavoidably stuck to the cool surface of the spinneret and thus broke the continuous spinning when they accumulated to a certain extent.

It should be noted that when the α-form cyclic dimer was mixed with water, caprolactam and other volatile matter, the sublimation could occur in advance at a temperature lower than 150°C and thus seriously interfere with the results.

Besides, we noted that very little of needles melt and left some liquid dots on the glass substrate; it is investigated more carefully by DSC and Schlenk tube test in following paragraphs.

## 3.5. Variable-temperature FT-IR spectroscopy

To investigate the transition between the α-form and β-form cyclic dimer, variable-temperature FT-IR spectroscopy was used to collect the signals of the crystals by varying the scan temperature from room temperature to 360°C. The curves are displayed in figure 7.

As shown as figure 7a, the stretching vibration of the amino groups (N–H) of the α-form shifts to the higher wavenumber with the increasing temperature, which is consistent with the weakening of bonds at higher temperatures and consistent with the previous report. Interestingly, the signals vanished as the

rsos.royalsocietypublishing.org   R. Soc. open sci. **5**: 180957

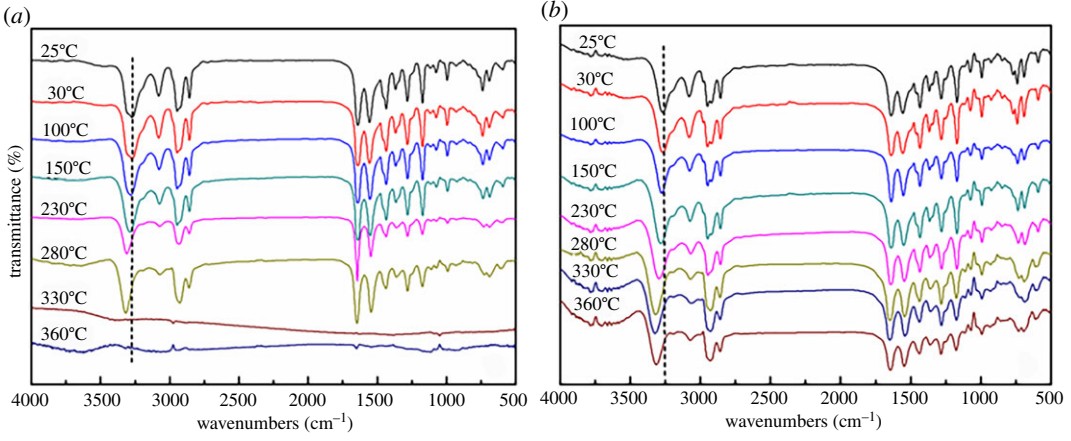

**Figure 7.** Variable-temperature FT-IR curves of cyclic dimer. (*a*) $\alpha$-form and (*b*) $\beta$-form.

temperature increased to over 330°C, which is explained by the sublimation of the cyclic dimer at a high temperature. Figure 7*b* indicates the FT-IR curves of $\beta$ crystals over all the temperature ranges, and it is clear that the shifting of the stretching vibration of the amino groups (N–H) in the $\beta$-form is more significant than that of the $\alpha$-form. Moreover, when the temperature is over 240°C, the two characteristic peaks at 2952 and 2921 cm$^{-1}$ combined into one appearing at approximately 2930 cm$^{-1}$, which shows that the $\beta$-form has begun to transform into the $\alpha$-form at this temperature. Additionally, the signals could be observed over the course of the heating programme, because the $\beta$-form is very stable before the transition temperature. It takes some time and heat energy to transform this dimer into the $\alpha$-form completely; hence, there are still enough $\alpha$ crystals remaining in the measuring stand during the latter stage of the rapid heating process.

## 3.6. Variable-temperature X-ray diffraction

Variable-temperature X-ray diffraction (XRD) has also been conducted to analyse the transition between the two forms of the cyclic dimer. As shown in figure 8*a*, the $2\theta$ curves for the $\alpha$-form crystal remained unchanged during the scanning process. This is very similar to that of the variable-temperature FT-IR spectroscopy.

However, the detection signals of the $\beta$-form crystals remain stable at lower temperatures and then exhibit a marked change over 230°C (figure 8*b*). The characteristic peaks of the $\beta$-form at 12.88°, 13.35°, 15.08° and 15.44° began to shrink. By further comparing the scanning curves at 240°C and 245°C, it is easy to identify the characteristic peaks at 11.48° attaching to the $\beta$-form, and a new peak at $2\theta$ of 10.85° is observed, which is attributed to the $\alpha$-form cyclic dimer. Along with the continuously increasing temperature to 250°C, the peak at 10.85° is more invisible, and the peaks for 11.48° shrank significantly in the curve. In the meantime, the signal of the other four characteristic peaks of the $\beta$-form at 12.88°, 13.35°, 15.08° and 15.44° also disappeared correspondingly, and another new peak belonging to the $\alpha$-form appeared at 15.39°. The transition was completed when the temperature reached over 255°C.

To investigate the initial and ending temperatures of the transition in more detail, the signals were collected from 240°C to 254°C with a temperature interval of 2°C, and the curves are shown in figure 8*c*. By comparing these scanning signals, we concluded that the transition occurred at the temperature of 244°C and was completed at 252°C.

In addition, because the transition from the $\beta$-form to the $\alpha$-form is a spontaneous endothermic process, the transformation speed of the cyclic dimer highly depends on the mass of the sample and the heating progress. With a lower quantity of sample and enough thermal energy supplied in a short time, it would be possible to accelerate the transformation of the $\beta$-form, or the transition time would extend and the mixed-signal of two types of cyclic dimer appear in the curve simultaneously at higher temperatures.

## 3.7. Thermal analysis

The thermal behaviours of the cyclic dimers were also analysed in this study. The heating curves obtained at a ramp rate of 10°C are presented in figure 9*a*. For the first and second scanning curves of the $\alpha$-form at the heating stage, there is only a peak appearing at approximately 345.44°C. When

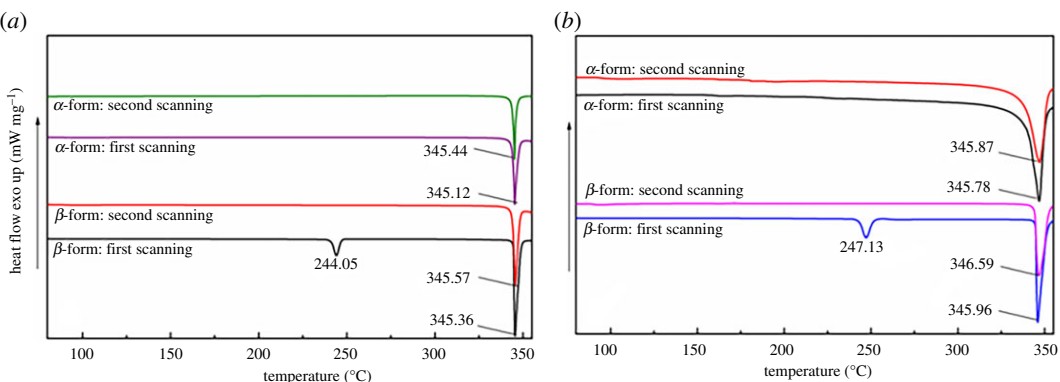

**Figure 8.** Variable-temperature XRD curves of cyclic dimer. (*a*) α-form; (*b,c*) β-form.

**Figure 9.** DSC patterns of cyclic dimer. (*a*) At a heating rate of 10°C; (*b*) at a heating rate of 20°C.

heating the sample from 50°C to 360°C again, one peak at 345.12°C was still observed, but the area of the melting peak shrank significantly because of the loss of sample over the course of heating. While two peaks can be observed in the first scanning curve of the β-form, the one at 244.05°C is attributed to the transition confirmed by FT-IR and XRD, and another that appears at approximately 345.36°C is regarded as the melting point of the α-form. Interestingly, the transition peak of 244.05°C vanished from the second heating curve, which could be evidence that the β-form has completely transformed into the α-form during the heating process.

Very similar results were also discovered when heating the α- and β-forms at a ramp rate of 20°C (figure 9*b*). For the α-form, the melting peak appears at 345.78°C and 345.87°C in the first and second heating curves, while for the β-form, there are two peaks of 247.13°C and 345.86°C in the first heating

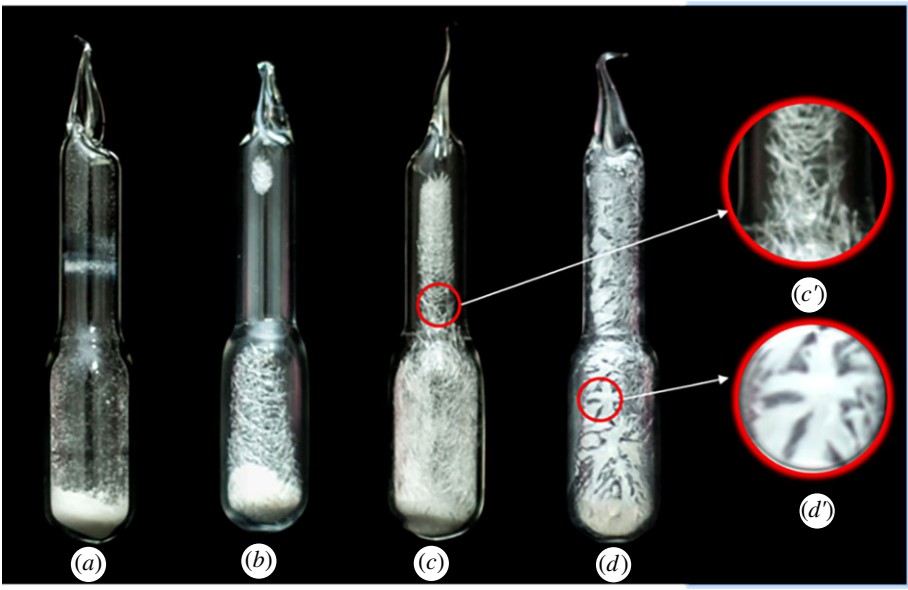

**Figure 10.** Sublimation, transition and melting of the $\beta$-form cyclic dimer. (*a*) $\beta$-form powder at room temperature; (*b*) heated at 250°C for 20 min; (*c*) heated at 320°C for 20 min; (*d*) heated at 360°C for 20 min; (*c'*) magnification of c circle zone; and (*d'*) magnification of d circle zone.

curve. After cooling the sample and heating it again, only one peak was observed at 346.59°C. Here, we noted that the transition peak of the $\beta$-form appeared at approximately 247.13°C, which was slightly higher than that obtained at the ramp rate of 10°C. Interestingly, the initial transformation temperature of the $\beta$-form measured at a heating rate of 20°C is 240.12°C, which is very close to the 239.98°C measured at the ramp rate of 10°C.

## 3.8. Sublimation

To investigate the sublimation and the melting behaviours that are more intuitive at high temperatures, Schlenk tubes containing the same amount (1.5 mg) of $\beta$-form powder were heated in a muffle furnace for 20 min at 250°C, 320°C and 360°C, respectively. Figure 10*a* shows the powder of the $\beta$-form at room temperature. By increasing the temperature to 250°C, bundles of needle-shape crystals were observed at the inner wall of the Schlenk tube as shown in figure 10*b*, which shows that some of the $\beta$-form has transitioned to the $\alpha$-form followed by sublimation. Figure 10*c* shows the morphology photographed at 320°C; interestingly, the Schlenk tube inner wall was fully covered by needles, and the evidence indicates that more of the $\beta$-form cyclic dimer transformed into the $\alpha$-form at a higher temperature. However, when the temperature increased to 360°C, we found that all the cyclic dimers melted into transparent liquid; some of them stayed on the bottom and some attached to the inner wall, where they solidified into stiff needles (figure 10*d*) very rapidly and stuck to the wall of the Schlenk tube with the falling temperature. This result is consistent with that observed by sublimating the cyclic dimer during the hot stage of polarized optical microscopy. The magnified pictures (*c'*, *d'*) of figure 9*c,d* show that the morphology of the needle crystals obtained at 360°C shows they are strongly bonded to one another, which makes them stiffer than those obtained at 320°C. To investigate the effect of the heating time and the mass on the transition and sublimation of the $\beta$-form in more detail, 1.5 g samples were heated at 250°C for 30, 40 and 60 min, respectively. The results showed that all the powder transitioned to the $\alpha$-form after being heated for 40 min. Additionally, by decreasing the mass of the sample to half, it only took 15 min for the $\beta$-form to transition completely to the $\alpha$-form.

## 4. Conclusion

In this study, the $\alpha$- and $\beta$-form cyclic dimers were prepared to high purity, and their characteristics and transition behaviours have been investigated in detail. Based on this research, we can conclude the following: (a) both forms could be fully dissolved in methanol and generate small regular rectangles

rsos.royalsocietypublishing.org R. Soc. open sci. 5: 180957

of the β-form upon solvent evaporation; and hot caprolactam solution is capable of dissolving a small amount of dimer but will form co-precipitates when the temperature is lower than 80°C. (b) The needle-like α-form cyclic dimer transitioned to the regular square β-form very quickly when encountering water, and it was easy to aggregate at high concentrations. (c) The β-form experienced three state changes during heating; at temperatures of 242–254°C, the β-form transitioned to the α-form slowly and then melted when the temperature increased over 345°C, or it quickly sublimated at a temperature higher than the transition temperature. (d) The β-form does not sublimate before the transition temperature, while the α-form cyclic dimer is a bit different; when the dimers were mixed with water, caprolactam and other volatile materials, they sublimated at temperatures lower than 150°C. (e) The transition of the cyclic dimer is strongly affected by the heating speed and the mass of the sample, especially for the α-form, and using much less sample sometimes leads to the disappearance of signals from the DSC and V-FT-IR scanning curves. Hence, it can be further concluded that during the whole manufacturing process of PA6 polymer, cyclic dimer exists only as β-form crystals in water-soluble extractables, concentrate and extracted PA6 chips, and brings serious problems to the process and quality of products. While in the spinning process, through melt-extruding at high temperature, the β-form transforms into needle-shape α-form again and might influence the continuous spinning process. Thus, many valuable works on the process optimization and quality improvement could be expected in the future.

Ethics. There is no necessity to complete an ethical assessment prior to our research. Our study does not involve any animals or humans.

Data accessibility. We mainly studied the physical properties of a cyclic dimer of PA6 in our paper, and these properties have been analysed by XRD, FT-IR, DSC, SEM and so on methods. The data and figures in the manuscript are very clear and enough to support the results reported. Hence, there are no additional supporting data for this paper.

Authors' contributions. L.P. and J.L. are two main implementers of the experiment and manuscript preparation. S.P. assisted the first two authors to complete the experimental data analysis. C.Y. guided the implementation of the entire research. While, F.J. made a great contribution to sample testing.

Competing interests. We have no competing interest.

Funding. This project was supported by the National Key Research Program of China (grant no. 2016YFB0302702) and Scientific Research Fund of Hunan Provincial Education Department (grant no. 17A126).

Acknowledgements. We are grateful for analytical support from Prof. Yumei Zhang of Donghua University. We are also grateful to Prof. Wenjie Liu for modifying the language of this manuscript.

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
