## [Reviewer comments · Royal Society Open Science]

Review History

RSOS-180957.R0 (Original submission)

Review form: Reviewer 1

Is the manuscript scientifically sound in its present form?

Yes

Are the interpretations and conclusions justified by the results?

Yes

Is the language acceptable?

No

Is it clear how to access all supporting data?

Yes

Do you have any ethical concerns with this paper?

No

Have you any concerns about statistical analyses in this paper?

I do not feel qualified to assess the statistics

Recommendation?

Reject

Comments to the Author(s)

Peng et al. describe a polymorphic transition of “cyclic dimers” using a range of analytical methods. I cannot recommend publication of the manuscript because I don’t feel that it brings anything new to the table, besides a lot of data. This data has not been used to learn anything new about polymorphism, nor it is explained how the data will affect polyamide-6 research and development. This aside, the manuscript is also not well written. In addition, the title is absolutely inappropriate because of the use of the word “miraculous”. I do not see anything miraculous about the described polymorphs and their transition. Not elucidating the mechanism does not make it miraculous. The authors are also encouraged to use appropriate chemical language. For example, the “cyclic dimer” should be described using IUPAC nomenclature. The manuscript may be suitable for publication after significant revisions. But at the moment I recommend a rejection. The required changes are too significant to justify a major revision.

Review form: Reviewer 2 (Chaosheng Wang)**Is the manuscript scientifically sound in its present form?**

Yes

Are the interpretations and conclusions justified by the results?

Yes

Is the language acceptable?

Yes

Is it clear how to access all supporting data?

Yes

Do you have any ethical concerns with this paper?

No

Have you any concerns about statistical analyses in this paper?

No

Recommendation?

Accept with minor revision (please list in comments)

Comments to the Author(s)

This paper seems very interesting in investigating the characteristics of PA6, particularly, it is the first time to show the whole transition process between α and β forms of cyclic dimer by using beautiful PLM pictures. Moreover, the influence of temperature on the change of α and β forms cyclic dimer is very important for readers to know and further study them. This paper presented enough experiment data and figures and has been well written. Hence I recommend it to be published in Royal Society Open Science after a minor revision.

1. Heading of Fig. 4 (Morphologies of cyclic dimer in different solutions: a: in methanol($\times 80$); b: in caprolactam($\times 80$); c: in water($\times 80$); d: aggregates in water($\times 80$); e: slice layers of aggregates (SEM $\times 500$); f: compact structure of aggregates (SEM $\times 500$)). The magnification should be shown on images by using scale, so that readers can understand the size of crystals more clear.
2. There is Chinese typing appeared in Figure 6 "Transition of cyclic dimer at heating program", authors should use new image to replace it.
3. Figure 10. Sublimating, transition and melting of β -form cyclic dimer: (d) magnification of d circle zone. Here (d) should be (d').

Decision letter (RSOS-180957.R0)

29-Aug-2018

Dear Mr Yi:

Title: Miraculous Transition Behaviors and Morphology Change of Polyamide 6 Cyclic Dimer
Manuscript ID: RSOS-180957

The editor assigned to your manuscript has now received comments from reviewers. We would like you to revise your paper in accordance with the referee and Subject Editor suggestions which can be found below (not including confidential reports to the Editor). Please note this decision does not guarantee eventual acceptance.

Please submit your revised paper before 21-Sep-2018. Please note that the revision deadline will expire at 00.00am on this date. If we do not hear from you within this time then it will be assumed that the paper has been withdrawn. In exceptional circumstances, extensions may be possible if agreed with the Editorial Office in advance. We do not allow multiple rounds of revision so we urge you to make every effort to fully address all of the comments at this stage. If deemed necessary by the Editors, your manuscript will be sent back to one or more of the original reviewers for assessment. If the original reviewers are not available we may invite new reviewers.

Once again, thank you for submitting your manuscript to Royal Society Open Science and I look

forward to receiving your revision. If you have any questions at all, please do not hesitate to get in touch.

Yours sincerely,
Dr Laura Smith, MRSC
Publishing Editor, Journals
Royal Society of Chemistry,
Thomas Graham House,
Science Park, Milton Road,
Cambridge, CB4 0WF, UK

Royal Society Open Science - Chemistry Editorial Office

On behalf of the Subject Editor Professor Anthony Stace and the Associate Editor Professor Claire Carmalt.

RSC Associate Editor:
Comments to the Author:
(There are no comments.)

RSC Subject Editor:
Comments to the Author:
Can the authors please find an alternative to "Miraculous" in the title. The journal does not deal in miracles!

Reviewers' Comments to Author:
Reviewer: 1

Comments to the Author(s)
Peng et al. describe a polymorphic transition of "cyclic dimers" using a range of analytical methods. I cannot recommend publication of the manuscript because I don't feel that it brings anything new to the table, besides a lot of data. This data has not been used to learn anything new about polymorphism, nor it is explained how the data will affect polyamide-6 research and development. This aside, the manuscript is also not well written. In addition, the title is absolutely inappropriate because of the use of the word "miraculous". I do not see anything miraculous about the described polymorphs and their transition. Not elucidating the mechanism does not make it miraculous. The authors are also encouraged to use appropriate chemical language. For example, the "cyclic dimer" should be described using IUPAC nomenclature. The manuscript may be suitable for publication after significant revisions. But at the moment I recommend a rejection. The required changes are too significant to justify a major revision.

Reviewer: 2

Comments to the Author(s)
This paper seems very interesting in investigating the characteristics of PA6, particularly, it is the first time to show the whole transition process between α and β forms of cyclic dimer by using beautiful PLM pictures. Moreover, the influence of temperature on the change of α and β forms cyclic dimer is very important for readers to know and further study them. This paper presented

enough experiment data and figures and has been well written. Hence I recommend it to be published in Royal Society Open Science after a minor revision.

1. Heading of Fig. 4 (Morphologies of cyclic dimer in different solutions: a: in methanol($\times 80$); b: in caprolactam($\times 80$); c: in water($\times 80$); d: aggregates in water($\times 80$); e: slice layers of aggregates (SEM $\times 500$); f: compact structure of aggregates (SEM $\times 500$)). The magnification should be shown on images by using scale, so that readers can understand the size of crystals more clear.
2. There is Chinese typing appeared in Figure 6 "Transition of cyclic dimer at heating program", authors should use new image to replace it.
3. Figure 10. Sublimating, transition and melting of β -form cyclic dimer: (d) magnification of d circle zone. Here (d) should be (d').

Author's Response to Decision Letter for (RSOS-180957.R0)

See Appendix A.

Decision letter (RSOS-180957.R1)

03-Oct-2018

Dear Mr Yi:

Title: The Crystal-form Transition Behaviours and Morphology Changes in a Polyamide 6 Cyclic Dimer

Manuscript ID: RSOS-180957.R1

It is a pleasure to accept your manuscript in its current form for publication in Royal Society Open Science. The chemistry content of Royal Society Open Science is published in collaboration with the Royal Society of Chemistry.

Yours sincerely,
Dr Laura Smith, MRSC
Publishing Editor, Journals
Royal Society of Chemistry,
Thomas Graham House,
Science Park, Milton Road,
Cambridge, CB4 0WF, UK

Royal Society Open Science - Chemistry Editorial Office

On behalf of the Subject Editor Professor Anthony Stace and the Associate Editor Professor Claire Carmalt.

RSC Associate Editor
Comments to the Author:
(There are no comments.)

Reviewer(s)' Comments to Author:

Appendix A

Dear editor:

Many thanks for the prompt response from you and the reviewers. We have checked our manuscript carefully and corrected it in detail according to the suggestions. The point to point responses to the referees are listed as below:

RSC Subject Editor:

Comments to the Author:

Can the authors please find an alternative to "Miraculous" in the title. The journal does not deal in miracles!

Thanks for the kind suggestion, we have changed the title to **“The Crystal-form Transition Behaviours and Morphology Changes in a Polyamide 6 Cyclic Dimer”**

Reviewers' Comments to Author:

Reviewer: 1

Comments to the Author(s)

Peng et al. describe a polymorphic transition of “cyclic dimers” using a range of analytical methods. I cannot recommend publication of the manuscript because I don't feel that it brings anything new to the table, besides a lot of data. This data has not been used to learn anything new about polymorphism, nor it is explained how the data will affect polyamide-6 research and development.

Thanks for the kind comments, although there are some papers also studied some of our contents, including the morphologies of β -form and α -form of PA6 cyclic dimer. While in fact, before our work, nobody investigated the whole transition process and morphology change between β -form and α -form by using PLM. This is very important for the researchers to know why there is only β -form existed in the concentrate (also named water-soluble extractable with higher concentration) and PA6 chips being extracted, which always cause serious problems to PA6 plants and quality of products, and why α -form appears again during spinning process at temperature over 250 °C, which like to adhere to the surface of spinneret and break the yarn. **In the manuscript we actually have explained how the data and morphology transition affect polyamide-6 research and development:**

“It is generally known that cyclic dimmer is insoluble in water, Hermans has

proved that the solubility of cyclic dimer in water is about 0.1%. However, by adding 10 times of water on the α -form cyclic dimer, the needle-like crystals transfer to regular square β -form very quickly although they are not soluble in water, as Fig.5(c) shows. These crystals show bigger size than those formed in methanol and easily aggregate precipitate in water (Fig.5(d)). More detail analysis executed by SEM discover that these aggregates mainly are existed in states of multi-slice structure (Fig.5(e)) and compact structure (Fig.5(f)). As reported as Giuseppe, each cyclic dimer molecule is connected by four hydrogen bonds to four other moleculars, **we believe the exposed molecular of cyclic dimer slices could also link the other slices, caprolactam, other oligomers and even water through hydrogen bonding at relative low temperature, thus reconstructed the larger and thicker plates. The facts can reasonably explain that in PA6 plants, the concentrates with high quantity of cyclic dimer are of great harmfulness as plugging up pipeline and equipment.**

In order to explain more clearly, we added the description “Moreover, in industrial PA6 plants, especially in monomer recovery unit, the β -form cyclic dimer in concentrate (also named water-soluble extractable with higher concentration) can be easily aroused great aggregation in equipment and pipes.”

In introduction and “Moreover, the larger and thicker β -form cyclic dimer plates are difficult to be removed completely from PA6 in extracting process due to its poor solubility in water, as a result, the remained β -form cyclic dimer in PA6 would and bring quality and safety problems of food packages, medical glue, baby clothes, and so on medical and life-care products, because they can migrate slowly from the inner to the surface of polymer. Hence, detailed technology of the transition of cyclic dimer can greatly help us to take the appropriate solutions, and implement controls to ensure the security of PA6 product” in 4.3, page 5.

And we also added another description “These studies clearly disclose a series of phenomenon happened in PA6 spinning process: firstly, the remained β -form cyclic dimer in PA6 transited to α -form during melt-extrusion at temperature higher than 250 °C, and then volatilized when they released from the small holes of spinneret, later, they were quickly cooled to needle-shape α -form cyclic dimer when exposed to the air. These needles were unavoidably stuck to the cool surface of spinneret and thus broke the continuous spinning when they accumulated to a certain extent. Besides, we noticed that very little of needles melt and left some liquid dots on the glass substrate, it would be investigated more carefully by DSC and schlenk tube test in following paragraphs.” In 4.4, page 6.

And we also added new conclusion in the end of the conclusion: “Hence, it can be further concluded that during the whole manufacturing process of PA6

polymer, cyclic dimer exists only β -form crystals in water-soluble extractables, concentrate and extracted PA6 chips, and brings serious problems to the process and quality of products. While in the spinning process, through melt-extruding at high temperature, the β -form transforms into needle-shape α -form again and might influence the continuous spinning process. Thus, many valuable works on the process optimization and quality improvement could be expected in the future.”

This aside, the manuscript is also not well written. In addition, the title is absolutely inappropriate because of the use of the word “miraculous”. I do not see anything miraculous about the described polymorphs and their transition. Not elucidating the mechanism does not make it miraculous.

Many thanks for this suggestion, we have sent the manuscript to the language edition company to improve the writing English and changed the unsuitable title to “The Crystal-form Transition Behaviours and Morphology Changes in a Polyamide 6 Cyclic Dimer”.

The authors are also encouraged to use appropriate chemical language. For example, the “cyclic dimer” should be described using IUPAC nomenclature.

Thanks for your kind comment. In PA6 field, people habitually use “cyclic dimer” instead of its IUPAC nomenclature(1,8-Diazacyclotetradecane-2,9-dione) since it is a professional term, in particular, many process engineer in PA6 plants know what is “cyclic dimer” but dot understand its IUPAC nomenclature. Hence, almost all authors used “cyclic dimer” in their published papers so that the readers can understand it easily. In the revised manuscript, we have given its IUPAC nomenclature (1,8-Diazacyclotetradecane-2,9-dione) in the abstract but kept using “cyclic dimer” in other places.

Reviewer: 2

Comments to the Author(s)

This paper seems very interesting in investigating the characteristics of PA6, particularly, it is the first time to show the whole transition process between α and β forms of cyclic dimer by using beautiful PLM pictures. Moreover, the influence of temperature on the change of α and β forms cyclic dimer is very important for readers to know and further study them. This paper presented enough experiment data and figures and has been well written. Hence I recommend it to be published in Royal Society Open Science after a minor revision.

Many thanks for the kind suggestions, we have checked the manuscript very

carefully and corrected the mistakes one by one.

1. Heading of Fig. 4 (Morphologies of cyclic dimer in different solutions: a: in methanol($\times 80$); b: in caprolactam($\times 80$); c: in water($\times 80$); d: aggregates in water($\times 80$); e: slice layers of aggregates (SEM $\times 500$); f: compact structure of aggregates (SEM $\times 500$)). The magnification should be shown on images by using scale, so that readers can understand the size of crystals more clear.

Thanks for the valuable suggestion. We have added the scale in Fig.5 (actually Fig.4 should be Fig.5) in the revision edition and rewrote the heading, please see the revised manuscript.

2. There is Chinese typing appeared in Figure 6 “Transition of cyclic dimer at heating program”, authors should use new image to replace it.

We have used a clearer image to replace it, thank you very much

3. Figure 10. Sublimating, transition and melting of β -form cyclic dimer: (d) magnification of d circle zone. Here (d) should be (d’).

Yes, this is a typing mistake and we have corrected it.

All other modifications were marked in red in the revised manuscript. We hope our modification can make you and the reviewers satisfied. Thanks again.

Yours sincerely:

Dr. Chunwang Yi

2018-9-14